# Influence of Molecular Design on the Tumor Targeting and Biodistribution of PSMA-Binding Tracers Labeled with Technetium-99m

**DOI:** 10.3390/ijms25073615

**Published:** 2024-03-23

**Authors:** Ekaterina Bezverkhniaia, Panagiotis Kanellopoulos, Ulrika Rosenström, Vladimir Tolmachev, Anna Orlova

**Affiliations:** 1Department of Medicinal Chemistry, Uppsala University, 751 23 Uppsala, Sweden; ekaterina.bezverkhniaia@ilk.uu.se (E.B.); panagiotis.kanellopoulos@ilk.uu.se (P.K.); ulrika.rosenstrom@ilk.uu.se (U.R.); 2Department of Immunology, Genetics and Pathology, Uppsala University, 752 37 Uppsala, Sweden; vladimir.tolmachev@igp.uu.se; 3Science for Life Laboratory, Uppsala University, 752 37 Uppsala, Sweden

**Keywords:** prostate cancer, prostate-specific membrane antigen, PSMA, technetium-99m, coupling linker, BQ0411, BQ0412, single-photon emission computed tomography

## Abstract

Previously, we designed the EuK-based PSMA ligand BQ0413 with an maE_3_ chelator for labeling with technetium-99m. It showed efficient tumor targeting, but our preclinical data and preliminary clinical results indicated that the renal excretion levels need to be decreased. We hypothesized that this could be achieved by a decrease in the ligand’s total negative charge, achieved by substituting negatively charged glutamate residues in the chelator with glycine. The purpose of this study was to evaluate the tumor targeting and biodistribution of two new PSMA inhibitors, BQ0411 and BQ0412, compared to BQ0413. Conjugates were radiolabeled with Tc-99m and characterized in vitro, using PC3-pip cells, and in vivo, using NMRI and PC3-pip tumor-bearing mice. [^99m^Tc]Tc-BQ0411 and [^99m^Tc]Tc-BQ0412 demonstrated PSMA-specific binding to PC3-pip cells with picomolar affinity. The biodistribution pattern for the new conjugates was characterized by rapid excretion. The tumor uptake for [^99m^Tc]Tc-BQ0411 was 1.6-fold higher compared to [^99m^Tc]Tc-BQ0412 and [^99m^Tc]Tc-BQ0413. [^99m^Tc]Tc-BQ0413 has demonstrated predominantly renal excretion, while the new conjugates underwent both renal and hepatobiliary excretion. In this study, we have demonstrated that in such small targeting ligands as PSMA-binding EuK-based pseudopeptides, the structural blocks that do not participate in binding could have a crucial role in tumor targeting and biodistribution. The presence of a glycine-based coupling linker in BQ0411 and BQ0413 seems to optimize biodistribution. In conclusion, the substitution of amino acids in the chelating sequence is a promising method to alter the biodistribution of [^99m^Tc]Tc-labeled small-molecule PSMA inhibitors. Further improvement of the biodistribution properties of BQ0413 is needed.

## 1. Introduction

Prostate cancer (PCa) is still the most common cancer type among men. PCa is asymptomatic in the early stage of the disease, has different clinicopathological features and progression patterns, and is characterized by a large number of indolent types of cancer, which require no treatment intervention [1,2]. However, the pinpointing of tumors that are aggressive and lethal despite having low Gleason scores is a clinical challenge. In these cases, new tools are needed to answer the title question [3]. Therefore, it is critical to develop an approach for early detection, disease stratification, and prediction of treatment response. Significant progress has been made in PCa biomarker discovery, mainly due to advances in genomic technologies. The development of these assays has opened up new opportunities to improve PCa diagnosis, prognosis, and treatment [4].

Commonly used diagnostic markers to confirm PCa include serum marker prostate-specific antigens (PSAs), androgen receptors, and prostate-specific membrane antigens (PSMAs) [5]. The PSMA protein, first described in 1987, is also known as glutamate carboxypeptidase II and folate hydrolase 1. PSMA is a transmembrane glycoprotein enzyme found on the cell surface. The internalization process that allows endocytosis of bound proteins enables radiolabeled PSMA tracers to accumulate within the cell. There is a minimal expression of PSMA in normal prostate tissue, the kidneys, the duodenum, the salivary and lacrimal glands, the brain, and the intestines; however, it has been found to be highly overexpressed (up to 1000-fold compared to healthy tissues) in PCa. PSMA is thus an attractive target for both molecular imaging and endoradiotherapy of prostate cancer, and extensive efforts have been made to develop new PSMA-targeting agents [6,7].

The PSMA-targeting tracers are divided mainly into two groups: monoclonal antibodies and small-molecule inhibitors (urea-based, phosphorus-based, and thiol-based molecules) [8]. Indium-111 capromab pendetide ([^111^In]In-capromab, ProstaScint^®^) was the first monoclonal antibody against PSMA used in PCa immunoscintigraphy. However, [^111^In]In-capromab lacks sensitivity because it recognizes an intracellular epitope of PSMA; thereby, it targets only apoptotic/necrotic or damaged cells [9]. Unlike [^111^In]In-capromab, J591, an antibody against the extracellular domain of PSMA, has shown improved targeting of prostate cancer in clinical trials [10]. Although antibodies offer potential for tumor targeting, their effectiveness as diagnostic tools is limited by a long half-life and poor tumor penetrability, particularly for bone metastases [11]. One of the first preclinical imaging studies using small-molecule PSMA inhibitors was reported by Foss et al. in 2005 [12]. The first clinical theranostic approach with radioiodinated versions of PSMA inhibitor MIP-1095, for the treatment of patients with metastatic castration-resistant prostate cancer, was reported in 2014 [13]. Currently, three small-molecule PSMA inhibitors, [^68^Ga]Ga-PSMA-11, [^18^F]F-DCFPyL, and flotufolastat F-18, have been approved for PET imaging of prostate cancer by the Food and Drug Administration (FDA). On March 23, 2022, the FDA approved Pluvicto (lutetium-177 vipivotide tetraxetan, also known as [^177^Lu]Lu-PSMA-617) for the treatment of adult patients with PSMA-positive metastatic castration-resistant prostate cancer (mCRPC) [14]. The development of PSMA-617 was a result of the optimization of linker moieties between the Glu-urea-Lys–based (EuK-based) binding motif and the DOTA chelator [15]. Another PSMA ligand that has shown promise for the therapy of mCRCP, [^177^Lu]Lu-PSMA-I&T, is currently being explored in a multicenter, randomized prospective phase III trial [16].

PSMA scans are upcoming diagnostic modalities, which detect metastatic lesions that are missed by conventional imaging modalities such as computed tomography, ultrasound, and magnetic resonance [17]. Because of the large numbers of PCa patients as well as the limited availability of PET cameras, SPECT imaging is still used more than PET imaging worldwide. The reasons are the lower cost and wider availability of SPECT-suitable radionuclides. Moreover, the advances made in SPECT technology, such as introducing cadmium zinc telluride (CZT) SPECT cameras, have markedly boosted SPECT spatial resolution and sensitivity and provided accurate absolute tracer uptake quantification comparable to PET. One of the most commonly used SPECT radionuclides is Tc-99m, as it has favorable photon energy (140 keV), is relatively inexpensive, and is readily available. Its physical half-life (t_1/2_ = 6 h) is compatible with the kinetics of small-molecule PSMA inhibitors. Thus, several small-molecule PSMA inhibitors ([^99m^Tc]Tc-MIP-1404, [^99m^Tc]Tc-MIP-1428, [^99m^Tc]Tc-PSMA-T4) have been developed [18,19].

For EuK-based PSMA-targeted small molecules, such as PSMA-617, major concerns are raised regarding kidney and salivary and lacrimal gland toxicity following PSMA-targeted radionuclide therapy [20]. In terms of imaging of prostate cancer, elevated uptake in normal PSMA-expressing organs may lead to non-optimal target/background ratios, likely resulting in lower imaging contrast and the underestimation of oligometastases. The key condition for successful use of labeled compounds is the high sensitivity and specificity of tumor imaging. Thus, preventive strategies to lower uptake in normal PSMA-expressing organs are needed. Structure modifications might be one of these options.

Modifications in small-molecule PSMA tracers, including linker length and the addition of multiple negative charges, were found to alter the binding affinity of the PSMA ligands, resulting in improved targeting characteristics and reduced background signals [21]. The fact that the placement of multiple negative charges on the linker does not lead to a significant reduction in affinity points to a novel engineering pathway for PSMA tracers with low non-specific binding [22]. At the same time, the introduction of a positively charged entity has proven to be a successful strategy for reducing renal retention of [^177^Lu]Lu-Ibu-DAB-PSMA, while the introduction of negatively charged residues led to elevated kidney retention [23]. Thus, it is generally accepted that the overall charge and charge distributions in the tracer itself can affect the pharmacokinetic profile [24].

Previously, we designed the EuK-based PSMA ligand BQ0413 (Figure 1), which contains an optimized 2-napththyl-L-alanine and L-tyrosine linker, responsible for affinity enhancement, and a negatively charged mercaptoacetyl-triglutamate chelator (maE_3_) for labeling with technetium-99m [25].

It was demonstrated that the incorporation of an maE_3_ chelator provided the best affinity towards PSMA in a series of previously optimized ligands with the same linker [25]. BQ0413 has demonstrated efficient tumor targeting, and the major clearance was through glomerular filtration. At the same time, improved affinity led to an elevated uptake in normal PSMA-expressing organs, such as the kidneys, salivary glands, spleen, and lungs. It was demonstrated that an increase in the mass of injected BQ0413 (i.e., a decrease in specific activity of the tracer) leads to the saturation of uptake in normal PSMA-expressing tissues with a minor impact on tumor uptake and decreased overall background. This effect results in optimal imaging contrast within several hours after the injection [25]. This approach could be also utilized for clinically used EuK-based PSMA inhibitors to overcome limitations regarding treatment-related salivary gland toxicity and renal toxicity.

Another alternative to decrease activity uptake in normal PSMA-expressing tissues is the modulation of physicochemical characteristics of the PSMA-targeting ligand BQ0413 through structure modifications. Studies on the influence of a chelator’s composition on the biodistribution of affibody molecules suggested that the nature of the amino acids in a chelator might influence the excretion pathway and thus improve tumor detection [26]. Currently, the spacer and the chelator are the hot spots for modifications in the development of new EuK-based PSMA ligands. PSMA inhibition potency, cellular internalization, and imaging quality could be strongly influenced even by slight differences in the linker and chelator moieties [15,27]. The kidney uptake of radiopharmaceuticals is likely mediated by multiple mechanisms, depending on the nature of the targeting vector. For example, the renal brush border has been shown to be negatively charged; therefore, the overall charge of the radiopharmaceutical may play a key role in determining the rate of reabsorption into the renal proximal tubular cells [28]. In view of the results obtained with the blocking of charge-dependent receptors, it is generally accepted that the overall charge and charge distributions in the tracer itself can affect the degree of kidney retention. Therefore, the introduction or removal of charges, as well as the position of the charges in the tracer, is an approach that has been widely investigated [24]. It was reported that the charge of a few amino acids on the surface of the peptide might be manipulated without affecting the affinity to the target, giving favorable properties for imaging [22]. Earlier studies on a [^99m^Tc]Tc-labeled affibody molecule containing a mercaptoacetyl-triglycyl chelator (maG_3_) showed favorable tumor-targeting properties (high tumor-to-blood ratios and excellent binding specificity) of the conjugate together with reduced kidney uptake [29,30]. The authors speculate that the use of glycine-containing chelators enables relatively low renal uptake of radioactivity. Taken together, we have hypothesized that the incorporation of an maG3 chelator might decrease the uptake and retention of radioactivity in normal organs, including PSMA-expressing organs.

In this study, we aimed to investigate if the chemical nature of amino acid chains in mercaptoacetyl-based chelators and the presence/absence of a coupling linker would influence the biodistribution properties. For this purpose, we have designed and preclinically evaluated two new PSMA inhibitors, BQ0411 and BQ0412, based on the previously developed BQ0413 inhibitor (Figure 2). The peptides were synthesized with and without a glycine-based coupling linker between the spacer and maG3 chelator. The overall charge in [^99m^Tc]Tc-labeled BQ0411 and BQ0412 is −1 [31], which is less negative than BQ0413 (−4) [25].

## 2. Results and Discussion

Radionuclide diagnostics is based on the preferential accumulation of a radioactive substance in a tissue of interest in comparison with normal surrounding tissues. In order to develop more efficient PSMA-targeting agents with enhanced tumor uptake and minimized off-target uptake, it is necessary to consider different structural features. Previously, we developed and preclinically evaluated the promising radioligand BQ0413. It showed efficient tumor targeting, but our preclinical data and preliminary clinical results indicated that renal excretion levels need to be decreased. We hypothesized that this could be achieved by a decrease in total negative charge, substituting negatively charged glutamate residues in the chelating sequence with uncharged glycine. The optimization of excretion pathways and uptake in excretory organs is an important issue in the design of targeting conjugates for molecular imaging. In this study, we designed and preclinically evaluated two new PSMA inhibitors, BQ0411 and BQ0412.

According to HPLC analysis, the retention times for non-labeled BQ0411 and BQ0412 were 14.3 and 14.1 min, respectively. The purity of the non-labeled compounds was sufficient for radiolabeling (Figure 3A and Figure 4A). BQ0411 and BQ0412 were successfully labeled with Tc-99m through a single-step procedure with an average radiochemical yield of 98.2 ± 0.4% (n = 7) and 98.2 ± 0.7% (n = 5), respectively, according to iTLC results. Since radiochemical yields were over 98% and reduced hydrolyzed technetium colloid (RHT) was less than 5%, no further purification was performed for in vitro and in vivo studies. The specific activity was 20 MBq/μg. To cross-validate radio-iTLC data, radio-HPLC analysis was performed. According to radio-HPLC chromatograms, the purity of [^99m^Tc]Tc-BQ0411 and [^99m^Tc]Tc-BQ0412 was more than 90%, and the retention times were 15.4 and 15.7 min, respectively (Figure 3B and Figure 4B).

Both [^99m^Tc]Tc-BQ0411 and [^99m^Tc]Tc-BQ0412 were stable during incubation in PBS at room temperature. [^99m^Tc]Tc-BQ0411 was stable in the presence of a 300-fold molar excess of cysteine. On the contrary, [^99m^Tc]Tc-BQ0412 showed around 60% release of activity (Figure A1A,B and Figure A2A,B). Two radiolabeled products were detected with retention times of 6.2 (32.5% of activity) and 14.1 (25.5%) min. Based on the literature, it could be speculated that the peak with the shorter retention time represents the Tc–cysteine complex, while the second peak most likely represents the intermediate transchelation product or binding to plasma proteins [32,33]. The cysteine challenge was proposed as a surrogate for in vivo stability studies already in the 1990s and has been considered as a predictor for the in vivo stability of Tc(O) complexes with cysteine- and mercaptoacetyl-based chelators [34]. However, according to the in vivo biodistribution results presented below, low uptake values of [^99m^Tc]Tc-BQ0412 in the stomach and salivary glands indicate that there was neither release of free pertechnetate into blood circulation during renal catabolism nor transchelation of Tc-99m with blood plasma proteins. The main difference between the system used by Hnatowich and by us is the size of the tested radiolabeled biomolecules: peptides vs. full-length antibodies. The rapid excretion of small peptides could occur before the release of pertechnetate becomes visible in the in vivo activity distribution.

The octanol–water distribution coefficient showed similar logD values for all three tracers (−2.3 ± 0.1 for [^99m^Tc]Tc-BQ0411 and [^99m^Tc]Tc-BQ0412, and −2.5 ± 0.1 for [^99m^Tc]Tc-BQ0413). Thus, structure altering through coupling linker modification and incorporation of glycine into a mercaptoacetyl-containing chelator did not result in a shift in overall hydrophilicity.

The in vitro binding specificity of [^99m^Tc]Tc-BQ0411 and [^99m^Tc]Tc-BQ0412 was tested using a saturation experiment. The binding significantly (*p* ˂ 0.0001) decreased when the PSMA-expressing PC3-pip cells were pre-treated with the excess amount of non-labeled PSMA inhibitor PSMA-11. This demonstrated that the binding of [^99m^Tc]Tc-BQ0411 and [^99m^Tc]Tc-BQ0412 is PSMA-mediated (Figure 5).

The binding kinetics of [^99m^Tc]Tc-BQ0411 and [^99m^Tc]Tc-BQ0412 to PSMA were measured in real time on living PC3-pip cells and compared with [^99m^Tc]Tc-BQ0413. The sensorgrams are shown in Figure 6. For [^99m^Tc]Tc-BQ0411 and [^99m^Tc]Tc-BQ0412, the 1:2 interaction model provided the best fitting. Reasonably high on-rate and slow off-rate constants were observed for all the tested radiotracers (Table 1). No large differences in the equilibrium dissociation constant (K_D_) values for [^99m^Tc]Tc-BQ0411 and [^99m^Tc]Tc-BQ0412 were observed. K_D_1 was within the picomolar range and K_D_2 was within the low nanomolar range. This indicates that the presence/absence of a glycine-based coupling linker does not have a remarkable effect on the agents’ affinity. However, the reference compound [^99m^Tc]Tc-BQ0413 demonstrated the best K_D_-value and, thus, the strongest affinity, demonstrating that the replacement of glutamate by glycine in the mercaptoacetyl-containing chelator somewhat decreased affinity. However, the picomolar affinity of [^99m^Tc]Tc-BQ0411 and [^99m^Tc]Tc-BQ0412 was sufficient for in vivo targeting. The binding of [^99m^Tc]Tc-BQ0413 was fitted both with a 1:1 and 1:2 interaction model. In both cases, K_D_1 was within a low picomolar range (89 × 10^−12^), which was lower than the K_D_1 values for [^99m^Tc]Tc-BQ0411 and [^99m^Tc]Tc-BQ0412. According to the 1:2 interaction model, K_D_2 for [^99m^Tc]Tc-BQ0413 was 5.7 × 10^−18^, which might not be a reliable value due to the long residence time of the ligand and the extremely slow dissociation rate.

Long tumor retention and internalization of the ligand–receptor complex seem to be essential for the success of an imaging agent. The cellular processing of the radiolabeled PSMA binders (Figure 7A,B) showed increasing cell-associated activity up to 4 h of continuous incubation, which is in agreement with k_a_ values. Cellular uptake was characterized by relatively rapid internalization. The maximum of internalized activity reached 44 ± 1% for [^99m^Tc]Tc-BQ0411 and 43 ± 1% for [^99m^Tc]Tc-BQ0412 at 4 h of incubation. After 4 h, the internalized activity remained at the same level of cell-associated activity for [^99m^Tc]Tc-BQ0411 (44.2 ± 0.7%), while [^99m^Tc]Tc-BQ0412 demonstrated a decrease in cell-associated activity between 4 and 8 h of continuous incubation. The internalized activity of [^99m^Tc]Tc-BQ0412 dropped to 32 ± 1% of the cell-associated activity. The processing of [^99m^Tc]Tc-BQ0411 demonstrated stronger residualizing properties, similar to [^99m^Tc]Tc-BQ0413, compared to [^99m^Tc]Tc-BQ0412, which demonstrated features of a non-residualizing label. Since [^99m^Tc]Tc-BQ0411 and [^99m^Tc]Tc-BQ0412 have the same hydrophilicity (logD value of −2.3 ± 0.1), we exclude the possibility of an increase in non-specific adherence to the cell membrane. Thus, the presence/absence of a coupling linker resulted in different internalization profiles, which might be critical for enhanced tumor uptake.

Another parameter that is an important precondition for high-contrast imaging is low activity accumulation in normal tissue. This will lead to high imaging contrast. To investigate how the different coupling linkers and chelator structures would affect the pharmacokinetic properties, a biodistribution study of [^99m^Tc]Tc-BQ0411 and [^99m^Tc]Tc-BQ0412 was performed in NRMI mice 3 and 24 h pi, using [^99m^Tc]Tc-BQ0413 as a comparator (Table 2). At 3 h pi, there was a clear difference in biodistribution between the two tested tracers. For [^99m^Tc]Tc-BQ0411, the organs with the highest activity uptake were the kidneys, followed by the intestines with content. The uptake of [^99m^Tc]Tc-BQ0412 in the intestines with content was 1.6-fold higher, but the uptake in kidneys was 2.3-fold lower than for [^99m^Tc]Tc-BQ0411. [^99m^Tc]Tc-BQ0412 also demonstrated the highest hepatic uptake among the three tested tracers. Oppositely, [^99m^Tc]Tc-BQ0413 demonstrated the highest kidney uptake. Overall, the biodistribution pattern for the new conjugates was characterized by rapid excretion: the activity remaining in the body at 3 h pi was 40–50% from injected for [^99m^Tc]Tc-BQ0411 and 20–30% for [^99m^Tc]Tc-BQ0412 (excluding activity in the intestines with content). The difference in remaining activity was due to the higher activity excreted and accumulated in the kidneys for [^99m^Tc]Tc-BQ0411. While the previously developed [^99m^Tc]Tc-BQ0413 has demonstrated predominantly renal excretion with a high degree of activity reabsorption in the kidneys, the new conjugates underwent both renal and hepatobiliary excretion, which is not an optimal clearance for permitting the detection of abdominal metastases. A noticeably higher uptake of [^99m^Tc]Tc-BQ0413 in the kidneys, salivary glands, and spleen was expected due to physiological PSMA expression, which was reported previously for other PSMA-targeting agents [35]. Since [^99m^Tc]Tc-BQ0413 has demonstrated better affinity to PSMA, it shows higher activity retention in normal PSMA-expressing organs (the kidneys, salivary glands, and spleen) compared to [^99m^Tc]Tc-BQ0411 and [^99m^Tc]Tc-BQ0412. The activity uptake in the tested organs, except the excretory organs, did not exceed 3% IA/g at 3 h pi (Table 2).

At 24 h pi, the activity uptake in all organs had significantly decreased compared to 3 h pi for all three tested tracers. The highest activity uptake remained in the kidneys for [^99m^Tc]Tc-BQ0413 (77% washout), followed by [^99m^Tc]Tc-BQ0411 (97% washout) and [^99m^Tc]Tc-BQ0412 (98% washout). The results from the biodistribution study did not show any signs of instability for the tested tracers. Interestingly, [^99m^Tc]Tc-BQ0412, which showed release of activity under cysteine challenge in vitro, provided an adequately low activity uptake in the salivary glands and stomach (organs with a high uptake of free pertechnetate [36]), suggesting the absence of release of free technetium during catabolism in vivo. The observed difference in the excretory pathway in the biodistribution profile was not based on the hydrophilicity of the tested tracers, since they have demonstrated quite close logD values.

The tumor-targeting properties and biodistribution profile of [^99m^Tc]Tc-BQ0411, [^99m^Tc]Tc-BQ0412, and [^99m^Tc]Tc-BQ0413 were compared in PC3-pip tumor-bearing mice at 3 h pi (Table 3). Previously, we have demonstrated that the uptake in normal PSMA-expressing organs and tissues might be blocked by a decrease in specific activity with a minor effect on tumor uptake [25]. Thus, we used the same dose range as has been reported previously in order to obtain a favorable biodistribution profile and tumor targeting. Due to sufficient blocking of PSMA-mediated uptake in the kidneys, salivary glands, and spleen, there was no big difference in activity uptake in these organs between tracers in this experiment. However, we have observed significant differences in blood clearance, excretion pathways, and tumor uptake for the tested conjugates. Tumor uptake was 1.6-fold higher for [^99m^Tc]Tc-BQ0411 compared with [^99m^Tc]Tc-BQ0412 and [^99m^Tc]Tc-BQ0413. At the same time, [^99m^Tc]Tc-BQ0411 demonstrated higher activity concentration in blood compared to other tracers, which might indicate longer residence time for [^99m^Tc]Tc-BQ0411, and consequently resulted in higher tumor uptake. We could also not exclude that, for [^99m^Tc]Tc-BQ0413, the tumor uptake might be partially blocked compared with [^99m^Tc]Tc-BQ0411 and [^99m^Tc]Tc-BQ0412 due to better affinity, and that mild residualizing properties for [^99m^Tc]Tc-BQ0412, observed in the in vitro internalization assay, could decrease retention in the tumor. In agreement with the higher activity concentration in the blood, [^99m^Tc]Tc-BQ0411 demonstrated the highest uptake in the lungs and liver. The activity uptake in the gastrointestinal tract (with content) for both [^99m^Tc]Tc-BQ0411 and [^99m^Tc]Tc-BQ0412 was over 40%IA, despite moderate uptake in the liver, resembling the biodistribution pattern in NMRI mice. However, the activity uptake in the gastrointestinal tract was significantly lower for [^99m^Tc]Tc-BQ0411. Among the tested conjugates, [^99m^Tc]Tc-BQ0413 had the lowest uptake in the gastrointestinal tract.

High contrast is crucial for optimal imaging since it determines the diagnostic sensitivity. Tumor-to-organ ratios (T/O) are presented in Table 4. [^99m^Tc]Tc-BQ0411 demonstrated a significantly higher tumor-to-kidney ratio compared to [^99m^Tc]Tc-BQ0412. However, [^99m^Tc]Tc-BQ0413 exhibited more beneficial properties as an imaging agent due to having the highest tumor-to-blood, tumor-to-liver, tumor-to-small intestine, and tumor-to-muscle ratios compared to the new agents [^99m^Tc]Tc-BQ0411 and [^99m^Tc]Tc-BQ0412. Since there was no significant difference in kidney uptake among the tracers, [^99m^Tc]Tc-BQ0413 remains the most promising imaging probe due to its favorable pharmacokinetics.

Overall, the exchange of the maE_3_ chelator in the PSMA-targeting agent BQ0413 to an maG_3_ chelator showed a negative effect on binding affinity, although the picomolar affinity for [^99m^Tc]Tc-BQ0411 and [^99m^Tc]Tc-BQ0412 was sufficient for successful in vivo targeting. Based on the obtained results, it appears that amino acid substitution in the chelator significantly decreased the accumulation of activity in the kidneys. However, this was achieved at the cost of a shift from predominantly renal excretion to a combination of renal and hepatobiliary excretion for the new conjugates. This would restrict their utility for the detection of metastases in the abdominal area. One possible way to decrease hepatobiliary excretion is a substitution of the maG_3_ chelator with a polar maSSS (mercaptoacetyl-triseryl) chelator [26].

In this study, we have demonstrated that in such small targeting ligands as PSMA-binding EuK-based pseudopeptides, the structural blocks that do not participate in binding could have a crucial role in tumor targeting and biodistribution. The presence of a glycine-based coupling linker in BQ0411 and BQ0413 seems to be optimal in terms of biodistribution compared to BQ0412. [^99m^Tc]Tc-BQ0413 remains the most promising imaging probe for SPECT diagnostic imaging due to its favorable pharmacokinetics. An elevated kidney uptake of [^99m^Tc]Tc-BQ0413 could be considered as negative per se, but the possibility of blocking kidney uptake by increasing injected mass is positive. Moreover, since the position of the kidneys is well defined, the use of SPECT/CT makes it possible to distinguish the high radioactivity accumulation in the kidneys from other organs and tissues. In contrast, radioactivity in the gastrointestinal tract in the case of [^99m^Tc]Tc-BQ0411 and [^99m^Tc]Tc-BQ0412 is more problematic [37]. This can lead to false-positive findings because of the occasional formation of volumes with a high activity concentration in the content of the gastrointestinal tract.

## 3. Materials and Methods

BQ0411 and BQ0412 (structures are shown in Figure 2, and characterization is in Figure A3) were produced by Pepmic Co., Ltd. (Suzhou, China) according to our design. PSMA-11 was purchased from ABX Advanced Biochemical Compounds (Radeberg, Germany). The PSMA-transfected PC3-pip cell line was issued by Prof. Martin G. Pomper (Johns Hopkins University, Baltimore, MD, USA). The cells were cultured in RPMI-1640 media, and PC3-pip cells were maintained with the addition of 10 mg/mL of puromycin every second passage. Media supplements (fetal bovine serum, penicillin–streptomycin (100 IU/mL penicillin, 100 µg/mL streptomycin), 2 mM L-glutamine, and trypsin-EDTA solution for cell detachment were purchased from Biochrom AG (Berlin, Germany). Tc-99m was obtained as the pertechnetate by elution with an Ultra TechneKow generator (Mallinckrodt, Petten, The Netherlands) with sterile 0.9% sodium chloride (Mallinckrodt, Petten, The Netherlands). The activity was measured using an automated gamma spectrometer with a NaI(TI) detector (1480 Wizard, Wallac, Turku, Finland). For the formulation of the injection solution, activity was measured using a dose calibrator VDC-405 (Veenstra Instruments BV, Joure, The Netherlands) equipped with an ionization chamber.

### 3.1. Radiochemistry

BQ0412 and BQ0413 were dissolved in PBS. BQ0411 was dissolved in water/ACN/NH_4_OH 5% (3:1:1). The amount of ACN was within the range approved by the Food and Drug Administration (FDA) for residual solvents [38]. The concentration of NH_4_OH was within the range approved by the guidelines on water for pharmaceutical use [39]. Tc-99m labeling was performed through a single-step procedure using a lyophilized kit containing 5 mg of gluconic acid sodium salt, 75 µg of stannous chloride, and 100 µg of EDTA [25]. A total of 10 µg of BQ0411, BQ0412, or BQ0413 was added to a freeze-dried kit, followed by freshly eluted [^99m^Tc]Tc-pertechnetate (20 MBq per 1 µg), and the vial was incubated for 60 min at 90 °C.

The radiochemical yield (RCY) was determined using instant thin-layer chromatography (iTLC) strips (Agilent Technologies, Santa Clara, CA, USA) eluted with acetone (R_f_ = 0 for radiolabeled tracers and [^99m^Tc]Tc-colloid, R_f_ = 1 for [^99m^Tc]Tc-TcO_4_). The RHT in the mixture was determined using pyridine/acetic acid/water (10:6:3) as the mobile phase ([^99m^Tc]Tc-colloid: Rf = 1; other forms of Tc-99m and radiolabeled tracers: Rf = 0). iTLC was analyzed using the Cyclone Plus Storage Phosphor System (PerkinElmer, Waltham, MA, USA).

The iTLC data were cross-validated using radio-HPLC analysis, performed using a Hitachi Chromaster HPLC system with a radioactivity detector and Phenomenex Luna^®^ C18 column (100 Å; 150 × 4.6 mm; 5 µm) at room temperature (20 °C), in the following conditions: solvent A: 0.1% trifluoroacetic acid (TFA) in H_2_O, solvent B: 0.1% TFA in acetonitrile, flow rate: 1 mL/min. For identity and purity analysis of [^99m^Tc]Tc-BQ0411 and [^99m^Tc]Tc-BQ0412, the method with a gradient from 5 to 60% solvent B over 20 min was used.

To evaluate the chemical stability of the [^99m^Tc]Tc-maG_3_ complex, fractions of radiolabeled conjugates (10 µL, 0.25 µg) were incubated with a 300× molar excess of cysteine or with PBS at room temperature for 1 h. The test was run in triplicate.

### 3.2. Octanol/Water Distribution Coefficient

The LogD was determined experimentally based on the procedure described earlier [40]. Briefly, the radiolabeled compound of interest was added to a LoBind Eppendorf tube (Eppendorf, Hamburg, Germany) containing Milli-Q water (500 μL) and n-octanol (500 μL), and the mixture was vortexed and centrifuged. Thereafter, three fractions from each phase were collected and the activity was measured using a gamma counter. The LogD was calculated by dividing the average activity measured in octanol by the average activity measured in water.

### 3.3. In Vitro Characterization

For the in vitro binding specificity test, approximately 8 × 10^5^ of PC3-pip cells per well were seeded in 6-well plates 24 h before the experiment. Cells in three control wells were pre-saturated with a 250-fold excess of non-labeled PSMA-11 15 min before the addition of radiotracers, while the second set was treated with complete media. After 15 min of incubation at room temperature, [^99m^Tc]Tc-labeled radioligands were added to all wells to reach a concentration of 2 nM. After one hour of incubation at 37 °C, the cells were washed with PBS solution, detached, and collected. Cell-associated activity was measured using a gamma counter and presented as the percentage of added activity.

The binding kinetics of [^99m^Tc]Tc-labeled BQ0411, BQ0412, and BQ0413 were measured in real time using LigandTracer Yellow Instruments (Ridgeview Instruments AB, Uppsala, Sweden) on PC3-pip cells, as described earlier [25]. The activity binding was recorded at 5 nM on the radiolabeled tracers for 150 min. After measuring the binding kinetics, the medium containing the labeled tracer was replaced with fresh medium, and the dissociation was monitored over 7 h. The association rate (k_a_) and the dissociation rate (k_d_) were computed using a 1:1 kinetic binding model in TraceDrawer software version 1.9.2 (Ridgeview Instruments AB, Uppsala, Sweden), and the equilibrium dissociation constant K_D_ was calculated.

To evaluate the cellular processing of [^99m^Tc]Tc-labeled BQ0411 and BQ0412, PC3-pip cells were incubated with the radiolabeled tracer (2 nM), and at predetermined time points (1, 2, 4, and 8 h), cells were treated with 0.5 mL of 4 M urea solution in a 0.2 M glycine buffer, pH 2.5, for 5 min on ice. After that, the acidic supernatant was collected, and this fraction was considered to be the membrane-bound activity. To lyse the cells, 0.5 mL of 1 M sodium hydroxide solution was added and incubated at 37 °C for at least 20 min. The alkaline-containing solution was collected, and the basic fraction was considered to be the internalized activity.

### 3.4. In Vivo Assays

All in vivo experiments were carried out on mice purchased from Scanbur A/S (Sollentuna, Sweden). The in vivo studies were conducted according to the guidelines of the Declaration of Helsinki and approved by the Ethics Committee for Animal Research in Uppsala, Sweden, approval number: 5.8.18-00473/2021 (approved on 26 February 2021).

### 3.5. Biodistribution in NMRI Mice

Four NMRI mice were intravenously injected with 40 pmol of the radiotracer (60 kBq, the mass of the injected compound was adjusted with an unlabeled tracer). The mice were euthanized with a lethal intraperitoneal injection of ketamine/xylazine, followed by exsanguination at 3 and 24 h post-injection (pi). The organs of interest were collected, weighed, and measured for their activity content using a gamma counter. The uptake activity in organs was calculated as the percentage of injected activity per gram (%IA/g) for the blood, salivary glands, lungs, liver, spleen, stomach, and kidneys and as %IA for the remaining carcass and the rest of the intestines with content.

### 3.6. Biodistribution in Tumor-Bearing Mice

PSMA-expressing prostate cancer xenografts were implanted in BALB/c nu/nu mice by subcutaneous injection of PC3-pip cell suspension in PBS (10^7^ cells/animal in 100 µL). The average tumor weight was 0.3 ± 0.2 g at the time of the experiment. Four mice per data point were used. Mice were intravenously injected with 5 nmol of the radiotracer (60 kBq, the mass of the injected compound was adjusted with an unlabeled tracer). The mice were euthanized 3 h pi, and the samples were treated as described above.

### 3.7. Data Analysis

The obtained values are presented as the average with standard deviation. Data were assessed either by an unpaired, two-tailed *t*-test or by one-way ANOVA with Bonferroni correction for multiple comparisons using GraphPad Prism software version 10.00 for Windows (GraphPad Software, San Diego, CA, USA). The difference was considered significant when the *p*-value was less than 0.05.

## 4. Conclusions

The substitution of amino acids in the chelating sequence is a promising method for altering the biodistribution of [^99m^Tc]Tc-labeled small-molecule PSMA inhibitors. Particularly, the incorporation of a mercaptoacetyl-triglycine chelator resulted in a decrease in renal uptake compared to the mercaptoacetyl-triglutamate chelator, but it led to an undesirable increase in abdominal radioactivity, restricting the detection of metastases in the abdominal area. Further improvement of biodistribution properties is needed.

## Figures and Tables

**Figure 1 ijms-25-03615-f001:**
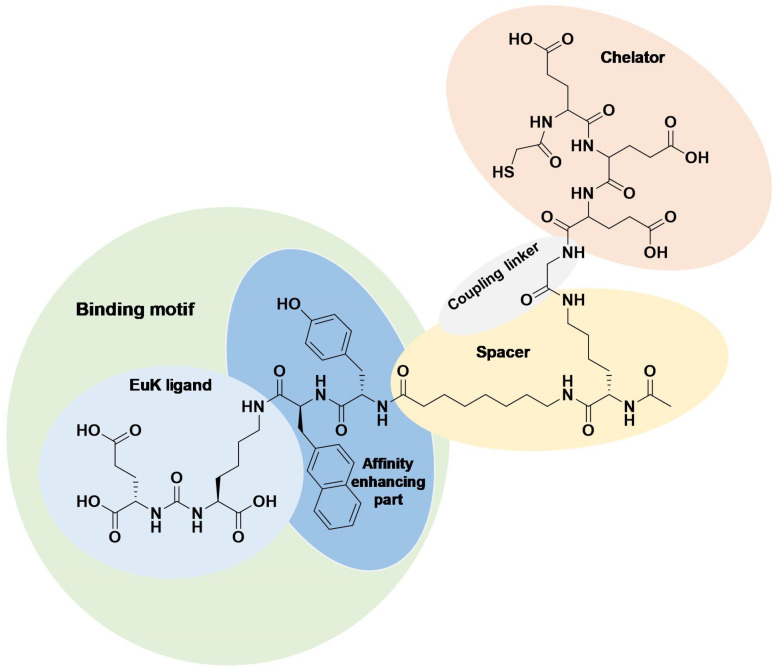
Chemical structure of BQ0413 with defined functional parts.

**Figure 2 ijms-25-03615-f002:**
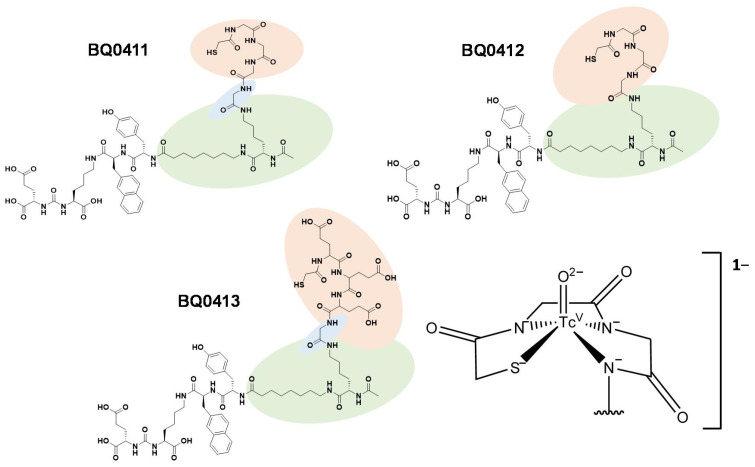
Chemical structures of BQ0411, BQ0412, and BQ0413, and the consensus structure of the Tc(O)maG3 complex [31].

**Figure 3 ijms-25-03615-f003:**
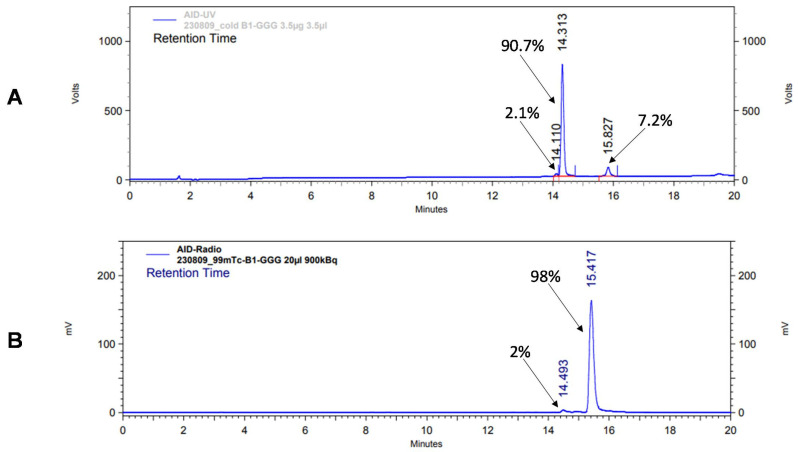
Reversed-phase HPLC chromatograms of non-labeled BQ0411 (**A**) and radiochromatogram of [^99m^Tc]Tc-BQ0411 (**B**). The retention times are expressed in minutes.

**Figure 4 ijms-25-03615-f004:**
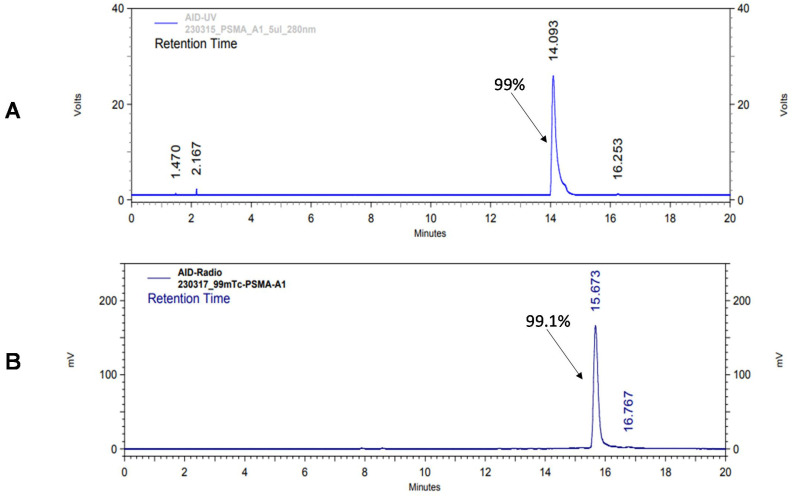
Reversed-phase HPLC chromatograms of non-labeled BQ0412 (**A**) and radiochromatogram of [^99m^Tc]Tc-BQ0412 (**B**). The retention times are expressed in minutes.

**Figure 5 ijms-25-03615-f005:**
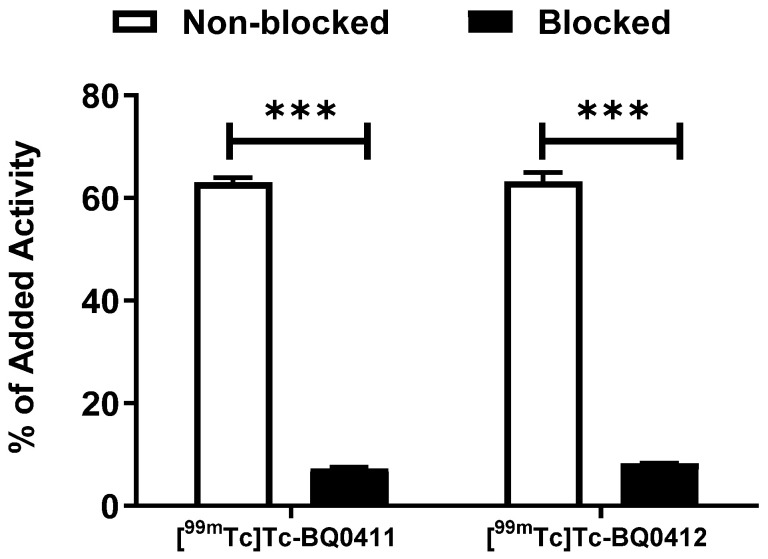
In vitro binding specificity of [^99m^Tc]Tc-BQ0411 and [^99m^Tc]Tc-BQ0412 on PC3-pip cell line. For pre-saturation of PSMA, a 250-fold molar excess of non-labeled PSMA-11 was added before adding labeled tracers. The error bars represent the standard deviation. *** indicates a *p*-value less than 0.0001 in an unpaired *t*-test.

**Figure 6 ijms-25-03615-f006:**
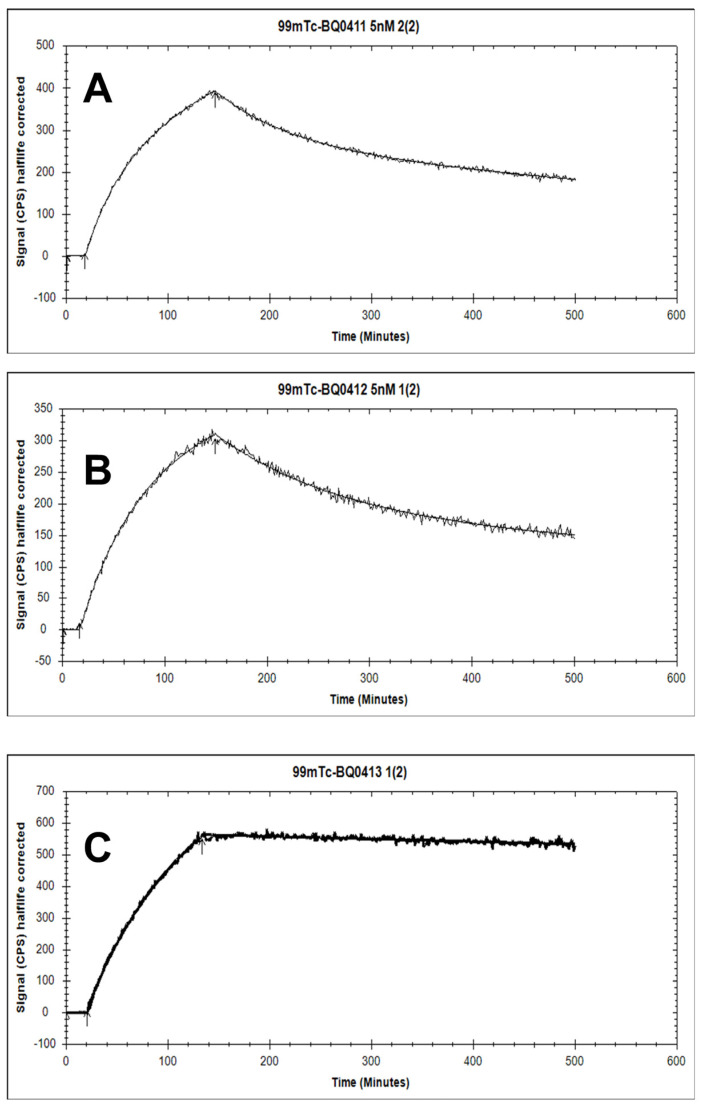
The sensorgrams obtained using LigandTracer Yellow for [^99m^Tc]Tc-BQ0411 (**A**), [^99m^Tc]Tc-BQ0412 (**B**), and [^99m^Tc]Tc-BQ0413 (**C**). The concentration during association measurements was 5 nM for each radiolabeled peptide.

**Figure 7 ijms-25-03615-f007:**
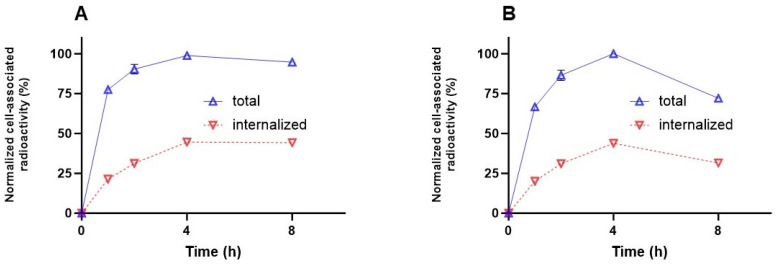
Cellular processing of [^99m^Tc]Tc-BQ0411 (**A**) and [^99m^Tc]Tc-BQ0412 (**B**) by PC3-pip cells during continuous incubation. Cells were incubated with 2 nM of each peptide at 37 °C. Data are presented as the mean of three samples ± standard deviation (SD). Error bars might not be seen when they are smaller than data point symbols.

**Table 1 ijms-25-03615-t001:** Affinity measurements of [^99m^Tc]Tc-BQ0411, [^99m^Tc]Tc-BQ0412, and [^99m^Tc]Tc-BQ0413.

Interaction Constants	[^99m^Tc]Tc-BQ0411	[^99m^Tc]Tc-BQ0412	[^99m^Tc]Tc-BQ0413
k_a_1 (M^−1^ s^−1^)	2.4 × 10^4^	3.1 × 10^4^	3.3 × 10^4^
k_d_1 (s^−1^)	1.4 × 10^−5^	1.4 × 10^−5^	2.9 × 10^−6^
K_D_1 (M)	604 × 10^−12^	440 × 10^−12^	89 × 10^−12^
k_a_2 (M^−1^ s^−1^)	6.7 × 10^4^	4.6 × 10^4^	2.9 × 10^8^
k_d_2 (s^−1^)	2.0 × 10^−5^	1.6 × 10^−5^	2.3 × 10^−9^
K_D_2 (M)	3.0 × 10^−9^	3.7 × 10^−9^	5.7 × 10^−18^

k_a_—association constant, k_d_—dissociation constant, K_D_—equilibrium dissociation constant.

**Table 2 ijms-25-03615-t002:** In vivo biodistribution of [^99m^Tc]Tc-BQ0411, [^99m^Tc]Tc-BQ0412, and [^99m^Tc]Tc-BQ0413 in NMRI mice at 3 and 24 h after injection of 40 pmol/mouse. Data are expressed as the percentage of administered activity (injected probe) per gram of tissue (%IA/g). Activity in intestines with content and carcass is presented as %IA/sample. The data are presented as the average (n = 4) and SD.

Organ	[^99m^Tc]Tc-BQ0411		[^99m^Tc]Tc-BQ0412		[^99m^Tc]Tc-BQ0413	
	3 h pi	24 h pi	3 h pi	24 h pi	3 h pi	24 h pi
Blood	1.4 ± 0.7 *^,a^	0.027 ± 0.002 ^b^	0.5 ± 0.1 *	0.016 ± 0.002 ^c^	0.5 ± 0.1 *	0.05 ± 0.01
Salivary	0.5 ± 0.1 *^,a,b^	0.022 ± 0.005 ^b^	0.24 ± 0.03 *^,c^	0.015 ± 0.006 ^c^	3.0 ± 0.1 *	0.30 ± 0.01
Lung	0.8 ± 0.1 *^,a^	0.019 ± 0.005 ^b^	0.3 ± 0.1 *^,c^	0.014 ± 0.008 ^c^	1.0 ± 0.3 *	0.18 ± 0.03
Liver	0.8 ± 0.1 *^,a^	0.032 ± 0.004 ^b^	3 ± 1 *^,c^	0.016 ± 0.004 ^c^	0.7 ± 0.1 *	0.21 ± 0.03
Spleen	0.38 ± 0.07 *^,b^	0.015 ± 0.002	0.23 ± 0.07 *^,c^	0.007 ± 0.005	1.2 ± 0.3 *	0.3 ± 0.2
Stomach	0.42 ± 0.07 *^,b^	0.06 ± 0.02 ^b^	0.17 ± 0.03 *^,c^	0.07 ± 0.05 ^c^	3.1 ± 0.5 *	1.0 ± 0.5
Kidney	129 ± 22 *^,a,b^	3.9 ± 0.6 ^b^	56 ± 9 *^,c^	1.1 ± 0.3 ^c^	242 ± 43 *	56 ± 14
Intestines with content	43 ± 6 *^,a,b^	0.3 ± 0.2	70 ± 7 *^,c^	0.16 ± 0.04 ^c^	3.8 ± 0.4 *	0.5 ± 0.2
Carcass	6.4 ± 0.8 *^,a,b^	0.21 ± 0.01 ^b^	2.9 ± 0.1 *^,c^	0.12 ± 0.03 ^c^	9 ± 1 *	1.4 ± 0.2

* Significant difference (*p* < 0.05 in 2-tailed *t*-test) between 3 h pi and 24 h pi for corresponding tracer; ^a^ significant difference (*p* < 0.05) between [^99m^Tc]Tc-BQ0411 and [^99m^Tc]Tc-BQ0412; ^b^ significant difference (*p* < 0.05) between [^99m^Tc]Tc-BQ0411 and [^99m^Tc]Tc-BQ0413; ^c^ significant difference (*p* < 0.05) between [^99m^Tc]Tc-BQ0412 and [^99m^Tc]Tc-BQ0413. ANOVA test (Bonferroni’s multiple comparisons test) was performed to test for significant (*p* < 0.05) differences.

**Table 3 ijms-25-03615-t003:** In vivo biodistribution of [^99m^Tc]Tc-BQ0411, [^99m^Tc]Tc-BQ0412, and [^99m^Tc]Tc-BQ0413 in PC3-pip tumor-bearing mice at 3 h after injection of 5 nmol/mouse. Data are expressed as the percentage of administered activity (injected probe) per gram of tissue (%IA/g). Activity in intestines with content and carcass is presented as %IA/sample. The data are presented as the average (n = 4) and SD.

Organ	[^99m^Tc]Tc-BQ0411	[^99m^Tc]Tc-BQ0412	[^99m^Tc]Tc-BQ0413
Blood	1.0 ± 0.4 ^b^	0.5 ± 0.1	0.17 ± 0.04
Salivary	0.4 ± 0.1	0.2 ± 0.1 ^c^	0.5 ± 0.1
Lung	0.9 ± 0.3 ^a,b^	0.5 ± 0.1	0.3 ± 0.1
Liver	1.9 ± 0.7 ^a,b^	0.5 ± 0.1	0.3 ± 0.1
Spleen	1.0 ± 0.2	0.6 ± 0.3	0.9 ± 0.5
Pancreas	0.2 ± 0.1	0.4 ± 0.3	0.13 ± 0.02
Stomach	0.4 ± 0.2	0.3 ± 0.2	0.6 ± 0.2
Small intestine	0.9 ± 0.4	1.2 ± 0.7 ^c^	0.2 ± 0.1
Kidney	13 ± 6	16 ± 5	9 ± 2
Tumor	27 ± 4 ^a,b^	17 ± 5	17 ± 3
Muscle	0.17 ± 0.05	0.2 ± 0.1	0.06 ± 0.02
Bone	0.20 ± 0.06	0.15 ± 0.07	0.2 ± 0.1
Intestines with content	44 ± 5 ^a,b^	69 ± 17 ^c^	1.7 ± 0.2
Carcass	3.5 ± 0.9	2.6 ± 1.3	1.7 ± 0.3

^a^ Significant difference (*p* < 0.05) between [^99m^Tc]Tc-BQ0411 and [^99m^Tc]Tc-BQ0412; ^b^ significant difference (*p* < 0.05) between [^99m^Tc]Tc-BQ0411 and [^99m^Tc]Tc-BQ0413; ^c^ significant difference (*p* < 0.05) between [^99m^Tc]Tc-BQ0412 and [^99m^Tc]Tc-BQ0413. ANOVA test (Bonferroni’s multiple comparisons test) was performed to test for significant (*p* < 0.05) differences.

**Table 4 ijms-25-03615-t004:** Tumor-to-organ ratios of [^99m^Tc]Tc-BQ0411, [^99m^Tc]Tc-BQ0412, and [^99m^Tc]Tc-BQ0413 in PC3-pip tumor-bearing mice at 3 h after injection of 5 nmol/mouse. The data are presented as the average (n = 4) and SD.

Organ	[^99m^Tc]Tc-BQ0411	[^99m^Tc]Tc-BQ0412	[^99m^Tc]Tc-BQ0413
Blood	32 ± 11 ^b^	35 ± 4 ^c^	94 ± 3
Salivary	69 ± 16 ^b^	90 ± 16 ^c^	31 ± 3
Lung	33 ± 9 ^b^	39 ± 6	52 ± 8
Liver	16 ± 7 ^a,b^	35 ± 5 ^c^	48 ± 5
Spleen	24 ± 7	26 ± 6	16 ± 4
Pancreas	121 ± 46	82 ± 33	137 ± 10
Stomach	86 ± 43 ^b^	77 ± 29	24 ± 2
Small intestine	35 ± 18 ^b^	19 ± 9 ^c^	126 ± 57
Kidney	2.3 ± 0.8 ^a^	1.1 ± 0.1	1.9 ± 0,3
Muscle	173 ± 47 ^b^	143 ± 58 ^c^	284 ± 19
Bone	146 ± 57	128 ± 38	114 ± 29

^a^ Significant difference (*p* < 0.05) between [^99m^Tc]Tc-BQ0411 and [^99m^Tc]Tc-BQ0412; ^b^ significant difference (*p* < 0.05) between [^99m^Tc]Tc-BQ0411 and [^99m^Tc]Tc-BQ0413; ^c^ significant difference (*p* < 0.05) between [^99m^Tc]Tc-BQ0412 and [^99m^Tc]Tc-BQ0413. ANOVA test (Bonferroni’s multiple comparisons test) was performed to test for significant (*p* < 0.05) differences.

## Data Availability

Data is contained within the article.

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
