# Peer review of "Influence of Molecular Design on the Tumor Targeting and Biodistribution of PSMA-Binding Tracers Labeled with Technetium-99m"

_ijms, 2024, doi:10.3390/ijms25073615_

Round 1

Reviewer 1 Report

Comments and Suggestions for Authors

The manuscript's title, "Influence of molecular design on the tumor targeting and biodistribution of PSMA-binding tracers labelled with technetium-99m," is relevant to the scope of the International Journal of Molecular Science and attracts attention. However, some revision and improvement are needed.

After analyzing the manuscript, I have some questions and suggestions for the authors listed below:

1.       How were the identities of BQ0411 and BQ0412 confirmed? Can you show any  MS or 1H, 13C-NMR analyses?

2.       The impurity of BQ0411 at a retention time of 15.827 min should be explained. What is the amount of these impurities in all compounds? Why does the pick of [99mTc]Tc-BQ0411 cover the retention time of impurity (Figure 3b)? Retention times and % AUC should be recorded on the chromatograms.

3.       Describe the picks that appear in Figure A2.

4.       How do the Authors explain the shift of approximately 1 min between the pick of unlabelled and labelled RF HPLS chromatograms?

5.       Lines 412-418: what is the Rf for [99mTc]Tc-colloid with acetone? Why do the Authors replace the PBS mobile phase used for [99mTc]Tc-BQ0413 [25; doi.org/10.3390/ ijms242417391] on pyridine: acetic acid: water for [99mTc]Tc-BQ0411 and [99mTc]Tc-BQ0412?

6.       Discussion about the T/O in Table 4 concerning tumour-to-liver ratios is untrue. It is lower in this case (16±7 compared to 35±5).

7.       In line 93, reference [19] describes [99mTc]Tc-PSMA-T4 as a developed compound, then [99mTc]Tc-PSMA-T1.

8.       Abbreviations should be defined the first time they appear in the text, e.g., an explanation of RHT on page 5, not on page 12.

9.       Lines 490 – 504 have to be removed.

10.   Description of references should be improved e.g. 16, 18,19.

Author Response

  1. How were the identities of BQ0411 and BQ0412 confirmed? Can you show any MS or 1H, 13C-NMR analyses?

Reply: Thank you for the valuable point. The HPLC and MS data for the products were added to the Appendix in revised version (note: the HPLC conditions were not identical to ones used for determination of yield and stability that is specified in Figure’s A3 legend).

  1. The impurity of BQ0411 at a retention time of 15.827 min should be explained. What is the amount of these impurities in all compounds? Why does the pick of [99mTc]Tc-BQ0411 cover the retention time of impurity (Figure 3b)? Retention times and % AUC should be recorded on the chromatograms.

Reply: Thank you for the valuable point. We have added the requested information to Figures 3 and 4. The impurity with retention time of 15.8 found in HPLC’ conditions for determination of yield and stability was below 10%, so the peptide product was considered pure enough for studies. However, it should be pointed that products had purity >95% (Figure A3. The nature of the impurity was not investigated, however this could reflect equilibrium between mono- and dianionic structure of Tc(O)MAG3 complex (Hansen et al. Structural characterization of the rhenium(V) oxo complex of mercaptoacetyltriglycine in its dianionic form. Metal-Based Drugs. 1995;2:105–110). Shifting of the BQ0411 peak to the longer retention time occurs due to the complexation of pertechnetate and some distance between UV and radio-detectors.

  1. Describe the picks that appear in Figure A2.

Reply: Thank you for the valuable point. The following text was added to Discussion:

                          Two radiolabeled product were detected with retention times 6.2 (29.6% of activity) and 14.1 (25.7%) min. Based on the literature it could be speculated that peak with the shorter retention time represents the Tc-cysteine complex, while the second peak most likely represents the intermediate transchelation product or binding to plasma pro-tiens [Rusckowski et al. A comparison in monkeys of (99m)Tc labeled to a peptide by 4 methods. J Nucl Med 2001; 42:1870 –1877, Vanderheyden et al. Evaluation of 99mTc-MAG3-annexin V: influence of the chelate on in vitro and in vivo properties in mice. Nucl Med Biol. 2006 Jan;33(1):135-44].

  1. How do the Authors explain the shift of approximately 1 min between the pick of unlabelled and labelled RF HPLS chromatograms?

Reply. Thank you for the valuable point. The shift of radiolabeled peaks in comparison with non-labeled peptides with high probability occurs due to the complexation of pertechnetate and some distance between UV and radio-detectors.

  1. Lines 412-418: what is the Rf for [99mTc]Tc-colloid with acetone? Why do the Authors replace the PBS mobile phase used for [99mTc]Tc-BQ0413 [25; doi.org/10.3390/ ijms242417391] on pyridine: acetic acid: water for [99mTc]Tc-BQ0411 and [99mTc]Tc-BQ0412?

Reply: Thank you for the question. Optimization of the mobile phase was performed within this project with the aim to get better discrimination between labeled peptide and free pertechnetate. The used acetone system is capable to discriminate between labeled peptide and free pertechitate, but not between labeled peptide and Tc-colloid. To clarify this issue to the readers we have added missed information to the text:

The radiochemical yield was determined using instant thin-layer chromatography (iTLC) strips (…) eluted with acetone (Rf = 0 for radiolabeled tracers and for [99mTc]Tc-colloid, Rf = 1 for [99mTc]Tc-TcO4). The reduced hydrolyzed technetium colloid (RHT) in the mixture was determined using pyridine:acetic acid:water (10:6:3) as the mobile phase ([99mTc]Tc-colloid: Rf = 1, other forms of Tc-99m and radiolabeled tracers: Rf = 0).

  1. Discussion about the T/O in Table 4 concerning tumour-to-liver ratios is untrue. It is lower in this case (16±7 compared to 35±5).

Reply: Thank you very much. We apologies for this mistake. The text on tumor-to-liver ratio was excluded.

  1. In line 93, reference [19] describes [99mTc]Tc-PSMA-T4 as a developed compound, then [99mTc]Tc-PSMA-T1.

Reply: Thank you, we have fixed this typo.

  1. Abbreviations should be defined the first time they appear in the text, e.g., an explanation of RHT on page 5, not on page 12.

Reply: We have moved the definition to p.5.

  1. Lines 490 – 504 have to be removed.

Reply: We apologies that text from template was left in manuscript.

  1. Description of references should be improved e.g. 16, 18,19.

Reply: The unnecessary additions generated by reference program were excluded.

Reviewer 2 Report

Comments and Suggestions for Authors

The manuscript describes the synthesis and pre-clinical evaluation of two new PSMA inhibitors for radiolabelling with Tc-99m. The two compounds differ by the presence of a glycine linker between the chelator and the targeting molecule. The triglycine chelators were also compared against an inhibitor with a triglutamate chelator that was previously described by the authors in Int. J. Mol. Sci. 2023, 24, 17391. Overall, the modifications did not result in improvements to the biodistribution. The manuscript investigates a concept that is well documented whereby changes to the chelator and the linker can modify biodistribution. In this case, the changes resulted in greater gastrointestinal uptake.  

Comments and suggestions for the authors:

How was the charge of the complexes determined?

Page 5. Definition of RHT should be provided when it first appears in the text. How was the RCY and RHT determined?

The coordination environment of the Tc ion should be provided in a figure and described. This could be playing a role in the biodistribution differences between the compounds.

Why use 300-fold molar excess of cysteine? Can identification of the resulting product be provided? Was there indication that the 60% Tc released from BQ0412 produced pertechnetate? Any discussion for why BQ0412 was less stable than BQ0411? Could it be due to the removal of the glycine?

Slightly higher lipophilicity for the new compounds according to logD values (-2.3 vs -2.5) is not considered significant. I would have thought that the difference would be sufficient to see differences.

How were the LigandTracer Yellow results analysed? What software?

Page 8 – How is ‘rapid excretion’ characterised? And in what time-frame? Bladder uptake?

Page 8, line 284-286. Indicate which are the ‘normal PSMA-expressing organs’ and which are the ‘healthy organs’ that aren’t excretory organs. Indicate the specific time points that relate to < ‘3% IA/g’. It is possible that this value was exceeded before 3 h p.i. or at some time point between 3 and 24 p.i. These values are snap shots of the activity uptake.

Line 291-294. What evidence is there that the product of the cysteine challenge is pertechnetate? No evidence of salivary gland and stomach uptake does not suggest that Tc-BQ0412 is unstable in vivo. Furthermore the difference in the excretory pathway could be due to instability and perhaps even due to the small difference in logD values. On the evidence presented, these possibilities can’t be ruled out.

Higher activity concentration in blood might indicate longer residence time which consequently resulted in higher tumour uptake – what ratio? (1.6 fold?)

Why is the highest concentration in the blood ‘in agreement’ with highest uptake in lungs and liver? Are these properties correlated?

Line 332 – is uptake in the gastrointestinal tract of Tc-BQ0412 in the gastrointestinal tract significantly lower compared to Tc-BQ0412? Provide statistical analysis

Line 385. Define ‘hot spots’

Comments on the Quality of English Language

See comments above

Author Response

  • How was the charge of the complexes determined?

Reply: The complex’s charge was dedicated based on the data published in lit literature, Kowalsky, Richard J. and Jeffrey P Norenberg. “Technetium Radiopharmaceutical Chemistry Continuing Education for Nuclear Pharmacists and Nuclear Medicine Professionals By.” (2006)..

Page 5. Definition of RHT should be provided when it first appears in the text. How was the RCY and RHT determined?

Reply: We have moved the definition to p.5. The determination of RCY and RHT are described in Section 3.1. Radiochemistry:

The radiochemical yield (RCY) was determined using instant thin-layer chroma-tography (iTLC) strips (…) eluted with acetone (Rf = 0 for radiolabeled tracers and for [99mTc]Tc-colloid, Rf = 1 for [99mTc]Tc-TcO4). The RHT in the mixture was determined using pyridine:acetic acid:water (10:6:3) as the mobile phase ([99mTc]Tc-colloid: Rf = 1, other forms of Tc-99m and radiolabeled tracers: Rf = 0). iTLC was analyzed using Cyclone Plus storage Phosphor System (…).

  • The coordination environment of the Tc ion should be provided in a figure and described. This could be playing a role in the biodistribution differences between the compounds.

Reply: Thank you very much for the relevant proposal. The dedicated complex structure was added to the Fig.2. The structure contains a distorted square pyramid, however it cannot be excluded that coordination spheres for BQ0411 and BQ0412 differ due to the presence of a glycine-based coupling linker in BQ0411.

  • Why use 300-fold molar excess of cysteine? Can identification of the resulting product be provided? Was there indication that the 60% Tc released from BQ0412 produced pertechnetate? Any discussion for why BQ0412 was less stable than BQ0411? Could it be due to the removal of the glycine?

Reply: The cysteine challenge was proposed by D.Hnatowich as a surrogate for in vivo stability studies already in 1990th (Hnatowich  et al. Can a cysteine challenge assay predict the in vivo behavior of 99mTc-labeled antibodies? Nucl Med Biol. 1994;21:1035-44) and was considered as a predictor for in vivo stability of Tc(O) complexes with mercaptoacetyl-based chelators. In our study, the in vitro stability test and in vivo biodistribution study were performed and parallel. The results of these two studies are antiparallel: while in vivo studies does not demonstrate evidence of complex instability (elevated activity uptake in salivary glands or stomach), we consider that cysteine challenging test is less relevant in our system. The main differences between the system used by Hnatowich and by us is a size of the tested radiolabeled biomolecules, peptides vs full length anbodies. The rapid excretion of small peptides could occur before the release of pertechnetate could to become visible in in vivo activity distribution.

We did not specifically performed identification of the products observed in in vitro stability test, but we can speculate that peak with the shorter retention time represents the Tc-cysteine complex, while the second peak most likely represents the intermediate transchelation product or binding to plasma proteins [Rusckowski et al. A comparison in monkeys of (99m)Tc labeled to a peptide by 4 methods. J Nucl Med 2001; 42:1870 –1877, Vanderheyden et al. Evaluation of 99mTc-MAG3-annexin V: influence of the chelate on in vitro and in vivo properties in mice. Nucl Med Biol. 2006 Jan;33(1):135-44].

We agree that presence/absence of glycine-based coupling linker might play a crucial role in the complex stability.

We have modified the paragraph on stability of labeled peptides as following:

Both [99mTc]Tc-BQ0411 and [99mTc]Tc-BQ0412 were stable during incubation in PBS at room temperature. [99mTc]Tc-BQ0411 was stable in presence of 300-fold molar excess of cysteine. On the contrary, [99mTc]Tc-BQ0412 showed 60% release of activity (Figure A1A,B. Figure A2A,B). Two radiolabeled product were detected with retention times 6.2 (29.6% of activity) and 14.1 (25.7%) min. Based on the literature it could be speculated that peak with the shorter retention time represents the Tc-cysteine complex, while the second peak most likely represents the intermediate transchelation product or binding to plasma proteins [Rusckowski ]. The cysteine challenge was proposed as a surrogate for in vivo stability studies already in 1990th and was considered as a predictor for in vivo stability of Tc(O) complexes with cysteine- and mercaptoacetyl-based chelators [Hnatowich]. However, according to in vivo biodistribution results presented below, low uptake values of [99mTc]Tc-BQ0412 in stomach and salivary gland indicate that there was neither release of free pertechnetate into blood circulation during renal catabolism, nor transchelation of Tc-99m with blood plasma proteins. The main differences between the system used by Hnatowich and by us is a size of the tested radiolabeled biomolecules, peptides vs full-length antibodies. The rapid excretion of small peptides could occur before the release of pertechnetate could to become visible in in vivo activity distribution.

  • Slightly higher lipophilicity for the new compounds according to logD values (-2.3 vs -2.5) is not considered significant. I would have thought that the difference would be sufficient to see differences.

Reply: We agree that despite very similar logD values we have observed sufficient differences in biodistribution of three tested peptide. The most evident one is different excretion pathway for the new variants: while Tc-BQ0413 demonstrated predominantly renal excretion with high degree of kidney reabsorption of excreted activity with minimal hepatobiliary excretion (both activity uptake in liver and content of GI tract were low 3 h pi), the new variants high degree of hepatobiliary excretion (3 h pi 43% and 70% of injected activity were accumulated in GI content for BQ0411 and BQ0412, respectively). However, BQ0411 and BQ0413 had identical LogD values.

  • How were the LigandTracer Yellow results analysed? What software?

Reply: The information is given in Section 3.3 In Vitro Characterization. Particularly, the association rate (ka) and the dissociation rate (kd) were computed using a 1:1 kinetic binding model in TraceDrawer software from Ridgeview Instruments AB, Uppsala, Sweden, and the equilibrium dissociation constant KD was calculated.

  • Page 8 – How is ‘rapid excretion’ characterised? And in what time-frame? Bladder uptake?

Reply: Thank you for the very interesting question. In the development of imaging agent, it is important that agent has rapid activity accumulation in targeted tissue(s), short blood circulation, preferably renal excretion to avoid diffused activity uptake in content of GI tract. There might be other requirements depending on the disease. For prostate cancer, low uptake in abdomen is desirable.  We considered “rapid excretion” as a characteristic of rapid blood clearance within the time comparable with imaging window for Tc-99m radiopharmaceutical in clinical setting (1 to 6 h pi), we have chosen 3 h pi in our experiments. We have observed that both new labeled peptides were cleared from the blood, with minimum uptake in non-targeting organs, we were able to estimate the degree of the hepatobiliary excretion based on activity accumulated in GI content (that was not emptied yet at this time point) and degree of activity “lost” via renal excretion with urine (as difference of injected activity and sum of activity in all samples). It is impossible to us to directly measure activity in urine due to national legislation on laboratory animals.

To address this issue we have added following text to Discussion:

Overall, the biodistribution pattern for the new conjugates was characterized by rapid excretion: the activity remained in body at 3 h pi was 40-50% from injected for [99mTc]Tc-BQ0411 and 20-30% for [99mTc]Tc-BQ0412 (excluding activity in intestines with content). The difference in remaining activity was due to higher activity excreted and accumulated in kidneys for [99mTc]Tc-BQ0411.

  • Page 8, line 284-286. Indicate which are the ‘normal PSMA-expressing organs’ and which are the ‘healthy organs’ that aren’t excretory organs. Indicate the specific time points that relate to < ‘3% IA/g’. It is possible that this value was exceeded before 3 h p.i. or at some time point between 3 and 24 p.i. These values are snap shots of the activity uptake.

Reply: We agree that measurements done at 3 h pi present a snap shot to the activity distribution at this time point. The required information was added to the text.

  • Line 291-294. What evidence is there that the product of the cysteine challenge is pertechnetate? No evidence of salivary gland and stomach uptake does not suggest that Tc-BQ0412 is unstable in vivo. Furthermore the difference in the excretory pathway could be due to instability and perhaps even due to the small difference in logD values. On the evidence presented, these possibilities can’t be ruled out.

Reply: Thank you for your question. Based on the literature we suggested that half of the products is Tc-cysteine complex and half is the intermediate transchelation product or binding to plasma proteins [Rusckowski ]. Our experience and literature data suggest that both salivery glands, stomach and thyroid do accumulate pertechnetate in the mechanism similar to accumulation of iodine via Na/I symporter [Jang Nuclear Medicine and Biology 110–111 (2022) 1; Liu et al. Pharmaceutics. 2022 May 20;14(5):1092]. This uptake can be blocked by addition of salts of iodide in the drinking water prio the experiment [Liu]. On the adequate stability of the new radioligands pointed their high uptake in PSMA expressing tumors (Table 3). Additionally, BQ0411 and BQ0413 had identical LogD values but very different excretion pathway.

  • Higher activity concentration in blood might indicate longer residence time which consequently resulted in higher tumour uptake – what ratio? (1.6 fold?) Why is the highest concentration in the blood ‘in agreement’ with highest uptake in lungs and liver? Are these properties correlated?

Reply: we agree that somewhat higher activity in blood for Tc-BQ0411 could be due to longer circulation in blood and could lead to higher uptake in targeted organs/tissues as was pointed by Reviewer. We did not perform accurate measurements of agents’ biological half-lives. The activity concentration in blood correlates with the activity uptake (concentration) in organs with high blood content, i.e. lungs, liver, spleen.    

  • Line 332 – is uptake in the gastrointestinal tract of Tc-BQ0412 in the gastrointestinal tract significantly lower compared to Tc-BQ0412? Provide statistical analysis

Reply: This information is provided in Table 3.

  • Line 385. Define ‘hot spots’

Reply: we agree with reviewer that this was not clear statement and reformulated this sentence:

This can lead to false-positive findings because of occasional formation of volumes with high activity concentration in content of gastrointestinal tract.